

# A simulation-based evaluation of methods for estimating census population size of terrestrial game species from genetically-identified parent-offspring pairs

Jeremy Larroque and Niko Balkenhol

Wildlife Sciences, University of Goettingen, Goettingen, Germany

## ABSTRACT

Estimates of wildlife population size are critical for conservation and management, but accurate estimates are difficult to obtain for many species. Several methods have recently been developed that estimate abundance using kinship relationships observed in genetic samples, particularly parent-offspring pairs. While these methods are similar to traditional Capture-Mark-Recapture, they do not need physical recapture, as individuals are considered recaptured if a sample contains one or more close relatives. This makes methods based on genetically-identified parent-offspring pairs particularly interesting for species for which releasing marked animals back into the population is not desirable or not possible (*e.g.*, harvested fish or game species). However, while these methods have successfully been applied in commercially important fish species, in the absence of life-history data, they are making several assumptions unlikely to be met for harvested terrestrial species. They assume that a sample contains only one generation of parents and one generation of juveniles of the year, while more than two generations can coexist in the hunting bags of long-lived species, or that the sampling probability is the same for each individual, an assumption that is violated when fecundity and/or survival depend on sex or other individual traits. In order to assess the usefulness of kin-based methods to estimate population sizes of terrestrial game species, we simulated population pedigrees of two different species with contrasting demographic strategies (wild boar and red deer), applied four different methods and compared the accuracy and precision of their estimates. We also performed a sensitivity analysis, simulating population pedigrees with varying fecundity characteristics and various levels of harvesting to identify optimal conditions of applicability of each method. We showed that all these methods reached the required levels of accuracy and precision to be effective in wildlife management under simulated circumstances (*i.e.*, for species within a given range of fecundity and for a given range of sampling intensity), while being robust to fecundity variation. Despite the potential usefulness of the methods for terrestrial game species, care is needed as several biases linked to hunting practices still need to be investigated (*e.g.*, when hunting bags are biased toward a particular group of individuals).

Corresponding author
Jeremy Larroque,
jeremy.larroque@uni-goettingen.de

## INTRODUCTION

For wildlife species that are hunted or fished, it is of a prime importance to identify the impact of harvesting on population dynamics (*Gosselin et al., 2015*). Monitoring the census population size is thus needed to assess whether or not the harvesting rate is sustainable (*Ratikainen et al., 2008*). It is also true for overabundant, pest or invasive species, for which biological control is mandatory (*Ratikainen et al., 2008*; *Sakai et al., 2001*).

Originally formulated by *Skaug (2001)*, the idea to use genetically-inferred kinship to estimate census population size (*i.e.,* the actual number of living individuals) of wild populations has recently gained attention and has been the subject of several methodological developments. These methods are similar to traditional Capture-Mark-Recapture methods (CMR, Box 1a) estimating census population size based on the recapture rates of marked individuals (*Schwarz & Seber, 1999*; *Seber, 1986*). Kin-based methods remove the need for physical marking and recapturing as individuals "mark" their relatives with shared genes (Box 1b; *Skaug, 2001*). Each individual can be physically trapped once, dead or alive, and considered recaptured if the sample contains one or more close relatives (*Skaug, 2001*). This makes these methods particularly interesting for species where releasing marked animals back into the population is not desirable or not possible, for example in invasive species or species of commercial harvest value (*e.g.,* fish or game).

To our knowledge, four different methods (Box 1b) have been developed to estimate census size from close kin data, particularly using parent–offspring pairs (POPs): the "Creel–Rosenblatt Estimator" (CRE, *Creel & Rosenblatt, 2013*), the "Close-Kin Mark-Recapture" (CKMR, *Bravington, Skaug & Anderson, 2016b*), the "Moment estimator" (*Hettiarachchige & Huggins, 2018*), and the "genetic-based Capture–Mark–Recapture" (g-CMR, *Müller, Mercker & Brün, 2020*). While based on the same principle, these methods differ in the kinship information required, the way they consider sampled and non-sampled individuals, and which population size they estimate (*i.e.,* full or adults' population size, both sexes or only females, Box 1b). To estimate adult population size, the CRE method (*Creel & Rosenblatt, 2013*) is based on the number of breeding individuals in the sample and the number of non-sampled individuals whose presence can be inferred by pedigree reconstruction (*i.e.,* individuals who bred with sampled mates and left offspring that were also sampled). CKMR (*Bravington, Skaug & Anderson, 2016b*) in its simplest "naïve" version only requires the identification of parent–offspring pairs in the sample to estimate adult breeding population size using an adaptation of the classical Lincoln–Petersen estimator (*Lincoln, 1930*; *Petersen, 1986*). The Moment estimator (*Hettiarachchige & Huggins, 2018*) requires the identification of mother-daughter pairs to estimate the number of breeding females in a population using the method of moments. Finally, the g-CMR method (*Müller, Mercker & Brün, 2020*) requires the identification of the number of father–offspring and mother–offspring pairs in the sample to estimate the total population

size also using an adaptation of the Lincoln–Petersen estimator (*Lincoln, 1930*; *Petersen, 1986*).

---

**Box 1.** Classic and kin-based capture-mark-recapture methods comparison

**(a) Capture-Mark-Recapture (CMR)**

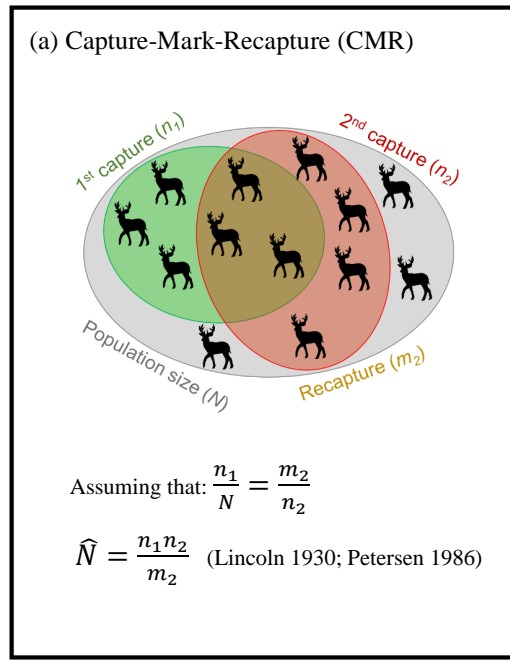

Assuming that: $\dfrac{n_1}{N} = \dfrac{m_2}{n_2}$

$$\widehat{N} = \frac{n_1 n_2}{m_2} \quad \text{(Lincoln 1930; Petersen 1986)}$$

(a) The simplest CMR model is the Lincoln-Petersen estimator which requires only two capture occasions. On the first occasion, a portion of the population ($n_1$) is captured, tagged, and released. The population is then re-sampled on one other occasion ($n_2$), and the ratio of marked ($m_2$) to unmarked animals is used to infer population size.

**(b) Kin-based CMR**

- Creel-Rosenblatt Estimator (CRE, Creel and Rosenblatt 2013)

$$\widehat{N}_A = n_s + 2n_{in} - \frac{n_{in}(n_F + n_M)}{n_s + n_{in}}$$

- Close-Kin Mark-Recapture* (CKMR, Bravington et al. 2016b)

$$\widehat{N}_A = \frac{2n_J n_A}{(n_{MO} + n_{FO})}$$

- Moment estimator (Hettiarachchige and Huggins 2018)

$$\widehat{N}_M = \frac{\hat{\mu} n_s (n_s - 1)}{n_{MD}(1 + \hat{\mu})^2}$$

- Genetic-based capture-mark-recapture (*g*-CMR, Müller et al. 2020)

$$\widehat{N} = n_J * \left( \frac{n_{AF}}{n_{MO}} + \frac{n_{AM}}{n_{FO}} \right) * \left( 1 + \frac{n_J}{(n_{AF} + n_{AM})} \right)$$

$\widehat{N}$, total population size
$\widehat{N}_A$, adults' population size
$\widehat{N}_M$, mothers' population size
$n_s$, number of individuals sampled
$n_{in}$, the number of individuals inferred by pedigree reconstruction
$n_F$, number of fathers sampled
$n_M$, number of mothers sampled
$n_A$, number of adults sampled
$n_J$, number of juveniles sampled
$n_{MD}$, number of mother-daughter pairs sampled
$\hat{\mu}$, mean number of daughters per mother
$n_{AF}$, number of adult females sampled
$n_{AM}$, number of adult males sampled
$n_{MO}$, number of mother-offspring pairs sampled
$n_{FO}$, number of father-offspring pairs sampled

(b) Kin-based CMR models assume that the presence of a close relative in the sample is equivalent to an individual recapture. The population is sampled once, individuals are genotyped, and these genotypes are used to determine the degree of genetic relatedness among individuals, *i.e.*, to identify close relative the sample, particularly parent-offspring pairs. Various information provided by the pedigree are then used to estimate population size. The CRE and CKMR methods estimate the adult population size ($\widehat{N}_A$). While CKMR only requires identifying mother- and father-offspring pairs ($n_{MO}$ and $n_{FO}$), CRE also requires determining the number of non-sampled individual but whose presence could be inferred ($n_{in}$), *i.e.*, individuals who bred with sampled mates and left offspring that were also sampled. The Moment estimator only requires the identification of mother-daughter pairs ($n_{MD}$) to estimates the number of breeding females ($\widehat{N}_M$) while the *g*-CMR method requires identifying $n_{MO}$ and $n_{FO}$ to estimate the total population size ($\widehat{N}$).

\* It should be noted that we presented here the simplest model of the CKMR method.

---

As evidenced by the numerous developments (*Anderson, 2022*; *Conn et al., 2020*; *Marcy-Quay et al., 2020*; *Patterson et al., 2022*; *Ruzzante et al., 2019*; *Waples & Feutry, 2022*), and practical applications (*e.g.*, *Bravington, Grewe & Davies, 2016a*; *Hillary et al., 2018*; *Prystupa et al., 2021*; *Trenkel et al., 2022b*) published recently, CKMR is currently the most utilized and best-studied method. Particularly, it can also incorporate life history information such as age or weight to account for difference in fecundity among age classes (*Bravington, Grewe & Davies, 2016a*). POPs-based methods, including the simplest model of the CKMR, assume that each individual in the population has the same probability of being captured

and recaptured, an assumption that is violated when fecundity and/or survival depend on sex or other individual traits. In practical applications of the CKMR method, researchers have accounted for this by adjusting the probabilities of capturing close kin based on sex-, age- or size-specific functions using a pseudo-likelihood approach, (*e.g., Ruzzante et al., 2019*). The data necessary to construct such correction functions are available for many commercially valuable fish species, where (tens of) thousands of individuals are harvested and sampled annually, and data on (sex- and age-/size-specific) fecundities and survival are routinely collected within mandatory monitoring programs (*e.g.*, for cod, *Ottersen et al., 2014*). For terrestrial game species, such data is rare, and we generally have to work with much smaller sample sizes. For example, in Germany the sex of harvested ungulates is usually reported, but the age is often only classified only into broad categories using tooth wear (*e.g.*, juveniles, yearlings, two-year old, *Wotschikowsky, 2010*), and especially for older individuals, the age estimation can also be quite inaccurate (*Hamlin et al., 2000*). For these reasons, we focused only on the simplest version of the CKMR method in this paper.

Many of the CKMR applications have been done in aquatic systems, to estimate the census sizes of different fish species or whales (*e.g., Bravington, Skaug & Anderson, 2016b; Trenkel et al., 2022a; Wacker et al., 2021*). Other methods have also been applied to terrestrial species, mostly those of conservation concern (*Creel & Rosenblatt, 2013; Hettiarachchige & Huggins, 2018; Spitzer et al., 2016*). However, estimating population size is also vital for sustainable management of terrestrial wildlife species that are hunted or trapped. While many methods for estimating population sizes in such species have been proposed, for example based on camera-traps (*e.g., Soofi et al., 2017*) and physical or genetic mark-recapture (*e.g., Ebert et al., 2012*), these methods require intensive fieldwork and/or expensive equipment. Hence, these methods are challenging to apply across large spatial scales and for long-term monitoring. In contrast, for species that are hunted, genetic samples of harvested individuals can be collected relatively easily and could provide the data necessary to apply POPs-based methods. Once sequenced, using microsatellites or single nucleotide polymorphisms (SNPs), individuals' data can be analyzed, looking for similarities between multilocus genotypes to infer the most likely genealogical relationships among individuals (*Blouin, 2003*) such as POPs. For example, identity-by-descent methods identify specific segments of DNA that are identical in two or more individuals, and use this information to estimate the probability that two individuals share a common ancestor within a certain number of generations (for a review of the different kinship estimation methods, see *Goudet, Kay & Weir, 2018*). This makes the idea of applying methods based on POPs particularly intriguing for terrestrial game species.

In this context, POPs-based methods using samples from harvested individuals could be a good alternative for estimating year-to-year population size and help to determine how harvesting impacts population growth, composition, and density. However, basic POPs-based methods assume that a sample contains only one generation of parents and one generation of juveniles of the year, meaning that individuals can be either parents or offspring, but not both. More than two generations can coexist for long-lived species, and hunting bags (*i.e.,* animals trapped or hunted during a given period) might contain individuals that are both offspring of older harvested individuals, as well as parents of

younger harvested individuals. It is not clear how individuals detected as both parent and offspring should be considered, and how their presence in the sample can bias the estimation.

In order to assess the usefulness of POPs-based methods to estimate population sizes of terrestrial game species, we simulated population pedigrees with known true population size, applied the four different methods, and compared the accuracy and precision of their estimations. For the purposes of this article, we restrict our discussion to comparing the performance of the different estimators, assuming perfect pedigree reconstruction. Our simulations mimicked two important game species in Europe with different demographic strategies, the red deer (*Cervus elaphus*) and the wild boar (*Sus scrofa*). While red deer females produce one or very rarely two offspring per year (*Clutton-Brock, 1985*), wild boar females can produce a higher and more variable number of offspring (*e.g.*, mean litter size = 6.6 [1, 12], *Frauendorf et al., 2016*). We hypothesized that because the pedigrees contain more than two generations, POPs-based methods would produce inaccurate population size estimates. We also hypothesized that the imprecision might be more pronounced for the wild boar, as the recapture rate will be more variable due to their more variable fecundity. Then, in order to identify optimal conditions for the applicability of each method, we performed a sensitivity analysis, simulating population pedigrees with varying fecundity characteristics. Finally, as populations can experience various levels of harvesting (*e.g.*, *Milner et al., 2006*), we also investigated the effect of sampling intensity, assuming that below a given threshold, estimation would become unreliable.

## MATERIAL AND METHODS

### Red deer and wild boar simulations

To simulate pedigrees, we used a custom R-script (Supplemental Information S1) and demographic parameters drawn from the red deer and wild boar literature. The simulation started with a given number of individuals whose age was randomly assigned between 1 and the maximum lifespan (Wild boar: 12 years, (*Jezierski, 1977*); Red deer: 15 years, *Lowe, 1969*), and whose sex was randomly attributed to obtain a sex-ratio of 0.5 (Red deer: *Bonenfant et al., 2003*; Wild boar: *Frauendorf et al., 2016*). For each generation, we then allowed sexually mature males and females to mate, males could reproduce with several females, while females could only mate with one male each year for the red deer, and with several males for the wild boar. Sexual maturity for the red deer was reached at 2 years-old for females (*Sibly et al., 2002*) and 5 years-old for males (age of social maturity, *Clutton-Brock & Albon, 1989*), while it was reached for the wild boar at 1 years-old for females (*Gethöffer, Sodeikat & Pohlmeyer, 2007*) and 3 years-old for males (*Brogi et al., 2021*). For the wild boar, 44% of the sexually mature females were allowed to mate with more than one male (*Gayet et al., 2016*), the number of partners was drawn from a Poisson distribution with in average $\lambda = 2$ partners (*Gayet et al., 2016*). The number of offspring per mother has been drawn from a Normal distribution with a mean fecundity of 1 ($\pm 0$) for the red deer (*Sibly et al., 2002*), and 4.9 ($\pm 2.1$) for the wild boar (*Fonseca et al., 2011*; *Náhlik & Sándor, 2003*). Yearling and newborn mortalities have been set up to 0.335 and 0.2 for the

red deer, and to 0.585 and 0.539 respectively for the wild boar, and a density-dependent adult mortality set up in average to 0.315 for the red deer (*Langvatn & Loison, 1999*) and to 0.360 for the wild boar (*Keuling et al., 2013*) to keep the population stable over time. For each species, we simulated 100 population replications of 500 individuals over 200 generations to let the population reach stability through the density-dependent adult mortality.

## Impact of species-specific fecundity and sampling intensity

We first compared the performance of the four methods for estimating red deer and wild boar population sizes. For this, we sampled 30% of the last generation in each simulated population 100 times to assess the accuracy and variability of the estimators. Second, to investigate the sensitivity of the four methods to fecundity, we simulated populations with varying fecundity characteristics: all the red deer demographic characteristics remained the same as described above, except that we first i) fixed the fecundity's standard deviation to 0 and varied the number of offspring per female from 1 to 14, and we then ii) fixed the fecundity to 7 and varied its standard deviation to three values (0, 2 and 4). These scenarios reflect a wide range of terrestrial game species. As before, we simulated 100 population replications of 500 individuals over 200 generations to let the population reach stability through the density-dependent adult' mortality and sampled the last generation. To determine the effect of the sampling intensity, for each replication of each scenario, we sampled the last generation 100 times with an intensity varying from 10 to 90% in increments of 10%. Finally, as the methods essentially depend on the identification of parent–offspring pairs (POPs), we also assessed results with respect to the actual number of POPs in the sample.

## Bias assessment

In order to determine the performance of the four methods (Supplemental Information S2), we applied the four estimators to population samples and then assessed their relative bias defined as the estimated census population size normalized by the true population size ($\hat{N}/N$). Relative bias is equal to 1 if the estimate matches the population size estimate exactly, <1 when the estimate is lower, and >1 if the estimate is higher than the actual population size. We assessed both bias accuracy (*i.e.,* mean value) and precision (*i.e.,* coefficient of variation [CV], presented as a percentage). For the g-CMR method, we estimated the whole population size (*Müller, Mercker & Brün, 2020*), for CRE and CKMR methods, we estimated the adult population size (*Bravington, Skaug & Anderson, 2016b*; *Creel & Rosenblatt, 2013*), and the breeding female population size was estimated for the Moment estimator (*Hettiarachchige & Huggins, 2018*). The CKMR, g-CMR and Moment methods do not assume overlapping generations, they consider POPs as adult-juvenile pairs only, and each individual can only be identified as parent or offspring. However, due to overlapping generations in the simulated populations, some individuals were identified both as parent and offspring, and the adult-juvenile distinction could therefore be no longer relevant. Thus, for the CKMR, g-CMR and Moment methods, we considered POPs between adults and juveniles, but also among adults.

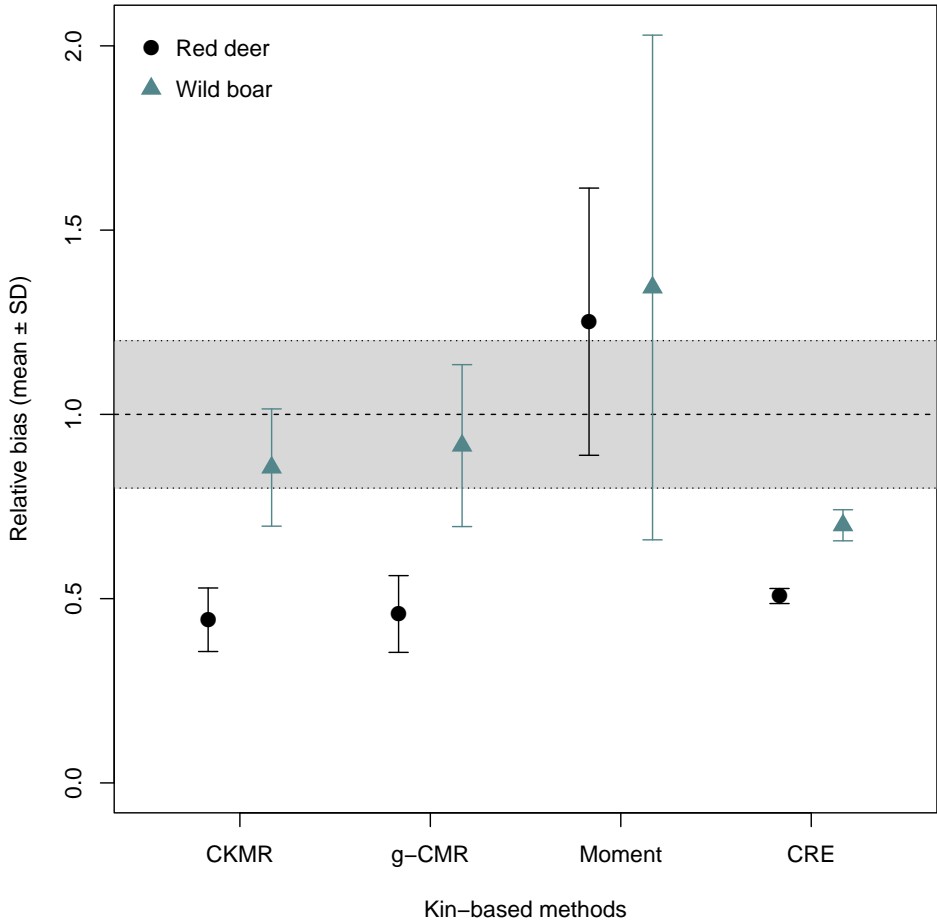

**Figure 1** **Relative bias ($\hat{N}/N$), mean ± SD) computed for the four kin-based capture-mark-recapture methods.** CKMR and CRE methods estimate adult population size while g-CMR and Moment methods estimate the whole population size and the breeding female population size, respectively. Bias is represented by black circles for the red deer and grey triangles for the wild boar. The long-dashed horizontal line represents the optimal value for an unbiased estimator, estimators above this line are overestimating the true population size while estimators below are underestimating it. Points in the grey area represent estimations within 20% of the true population size.

## RESULTS

When comparing the relative bias between the simulated wild boar and red deer, CKMR, g-CMR and CRE methods performed better for the wild boar than for the red deer (Fig. 1). Wild boar populations were underestimated by 14% on average for the CKMR method, by 8% for the g-CMR method, and by 30% for the CRE method, while red deer population sizes were always underestimated by ∼50%. The Moment estimator tended to overestimate population size for both species, but the bias and its variation were slightly lower for the red deer compared to the wild boar (25% *vs*. 34% on average, Fig. 1).

When investigating the effect of fecundity and sampling intensity, we found that the performance of CKMR, g-CMR, and Moment methods are most strongly linked to the
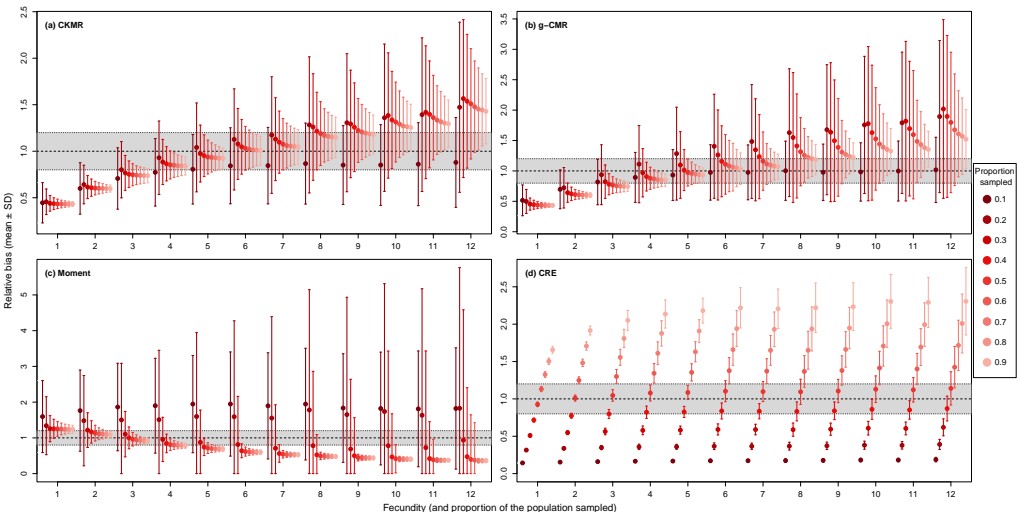

**Figure 2** **Relative bias ($\hat{N}/N$), mean ± SD) as a function of the fecundity (ranging from 1 to 12), and the proportion of the population sampled ranging from (0.1 to 0.9) for the (A) CKMR, (B) g-CMR, (C) Moment and (D) CRE methods.** CKMR and CRE methods estimate adult population size while g-CMR and Moment methods estimate the whole population size and the breeding female population size, respectively. The long-dashed horizontal line represents the optimal value for an unbiased estimator, estimators above this line are overestimating the true population size while estimators below are underestimating it. Points in the grey area represent estimations within 20% of the true population size. Results for fecundity > 12 are not shown to increase visibility (original figure can be found in Fig. S2).

mean fecundity value (Fig. 2). CKMR and g-CMR methods have similar performance with an optimal fecundity value ranging from 5 to 7 below which they tend to underestimate the population size by up to 50%, and above which they tend to overestimate it by up to 100% for the highest fecundity values. Bias reaches a plateau for fecundity values >12 (Figs. S1 and S3). The effect of fecundity is opposite for the Moment method, with an optimal fecundity value ranging from 2 to 4 offspring, below which population size tend to be overestimated, and above which it tends to be underestimated. Bias reached a plateau for fecundity values >8. For the CKMR and g-CMR methods, estimations for a sampling intensity of 10% are lower than for higher percentages and tend to remain the same whatever the fecundity values. It should be noted that for the g-CMR methods and for fecundity values >4, even a low sampling intensity of 10% leads to unbiased population size estimations. A bias reduction with increasing sampling intensity is also particularly noticeable for the g-CMR methods and for fecundity values >5. For the Moment method, estimations with a sampling intensity ≤ 30% tend to overestimate population size (up to 94%) compared to a sampling intensity ≥ 40% for which estimations remained stable. CRE method performance is clearly linked to the sampling intensity with an optimal value of 50%, below which population size is underestimated by up to 86%, and above which the size is overestimated whatever the fecundity value by up to 130%.

For the CKMR, g-CMR and Moment methods, the CV increased with the fecundity but decreased with the sampling intensity (Fig. 3). For the optimal fecundity values of the CKMR and g-CMR methods (*i.e.,* from 5 to 7, Fig. 2), only estimations based on a
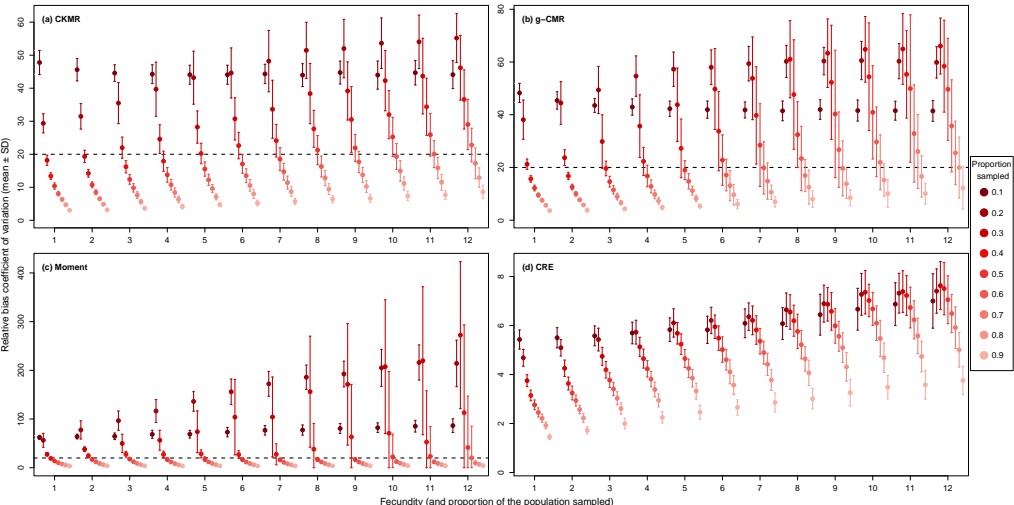

**Figure 3** Relative bias ($\hat{N}/N$) coefficient of variation (mean ± SD) as a function of the fecundity (ranging from 1 to 12), and the proportion of the population sampled ranging from (0.1 to 0.9) for the (A) CKMR, (B) g-CMR, (C) Moment and (D) CRE methods. CKMR and CRE methods estimate adult population size while g-CMR and Moment methods estimate the whole population size and the breeding female population size, respectively. The long-dashed horizontal line represents the optimal value of 20% for a precise estimator, estimators above this line are too variable to be useful for wildlife management and conservation. Results for fecundity > 12 are not shown to increase visibility (original figure can be found in Fig. S2).

sampling intensity >40% had a CV <20%, while the CV values for the lowest sampling intensity were up to 67%. The same phenomenon was observed for the optimal fecundity values of the Moment method (*i.e.,* from 2 to 4, Fig. 2) but the CV values for the lowest sampling intensity were up to 280%. The CRE method is the only one for which the CV was always lowest than 20% (Fig. 3). The fecundity variation had no effect (Fig. 4), as the same pattern depending only on the sampling intensity was found for the four methods for fecundity SD values tested.

While fecundity remains the most important factor, when based on less than 30 POPs, CKMR, g-CMR and Moment estimations are very variable, while estimations using ≥ 30 POPs show a clear reduction in their SD (Fig. 5). Above 60 POPs, there is no difference in the relative bias for the Moment method whatever the fecundity value, and until a fecundity of 8 offspring for the CKMR, and g-CMR methods. For fecundity values ≥ 8, increasing the number of POPs seems to decrease the bias. CRE optimal POPs value varies from 60 to 110 as a function of the fecundity.

## DISCUSSION

Our examination of POPs-based CMR methods to estimate population size shows their potential to inform game species conservation research and management, albeit with careful interpretation. Through simulation analyses, we have demonstrated that all these methods reached the required levels of accuracy and precision to be effective in wildlife
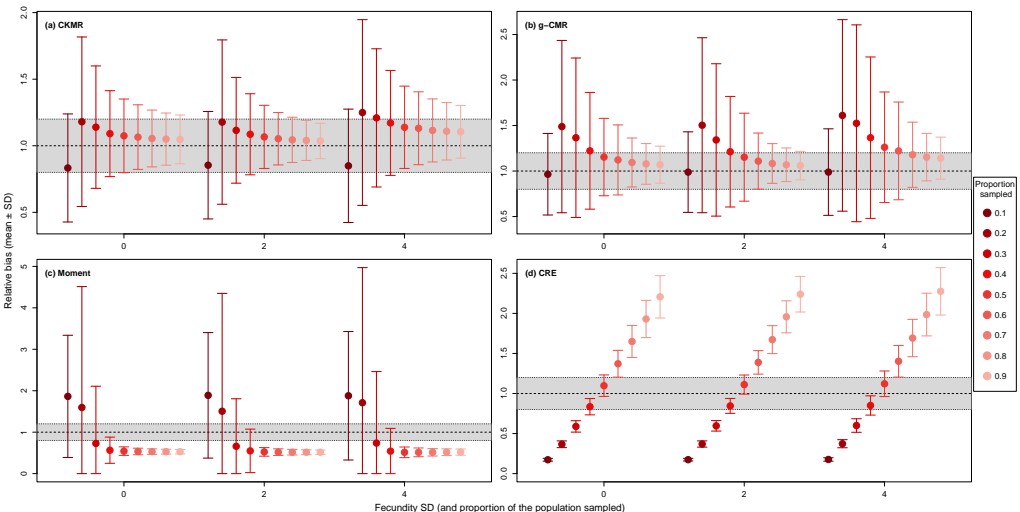

**Figure 4** Relative bias ($\hat{N}/N$), mean ± SD) for a fixed fecundity of 7 with an SD of 0, 2 and 4, and the proportion of the population sampled ranging from (0.1 to 0.9) for the (A) CKMR, (B) g-CMR, (C) Moment and (D) CRE methods. CKMR and CRE methods estimate adult population size while g-CMR and Moment methods estimate the whole population size and the breeding female population size, respectively. The long-dashed horizontal line represents the optimal value for an unbiased estimator, estimators above this line are overestimating the true population size while estimators below are underestimating it. Points in the grey area represent estimations within 20% of the true population size.

management (*i.e.*, CV ≤ 20%, *Pollock et al., 1990*) for species within a given range of fecundity and for a given range of sampling intensity.

As expected, sampling intensity was an important factor that must be considered carefully. For the CKMR, g-CMR and Moment methods with optimal fecundity values, a minimal sampling effort of 40% of the total population was required to reach an acceptable precision level (*i.e.*, CV ≤ 20%). This percentage is likely to be reached in certain hunted ungulate populations in Europe (*e.g.*, 32–63% for the wild boar in Europe, *Bassi et al., 2020*; *Merli et al., 2017*; *Toïgo et al., 2008*, 24% in average and up to 78% for the red deer, *Burbaite & Csányi, 2010*, 29% in average and up to 43% for the roe deer in Europe, *Burbaite & Csányi, 2009*), but less likely for carnivores species for which the density is much lower (*e.g.*, 17% max allowed for the wolf in France, *Meuret et al., 2020*). When the hunting pressure is too low, several other sample sources can be used to reach a higher sample size such as DNA collected from fecal samples (*e.g.*, *Prigioni et al., 2006*), snares (*e.g.*, *Gardner et al., 2010*) or roadkill (*e.g.*, *Allio et al., 2021*).

When based on less than 40% sampling intensity, estimations must be treated with caution as the CV can reach extreme values up to 280% for the Moment method for example. Arguably, biased estimates may be better than no estimates, for example when the goal is to track population changes over time rather than to estimate absolute population sizes. Nevertheless, estimations based on less than 40% sampling intensity should be treated with caution, particularly with such high CV, as bias might not be consistent over time,

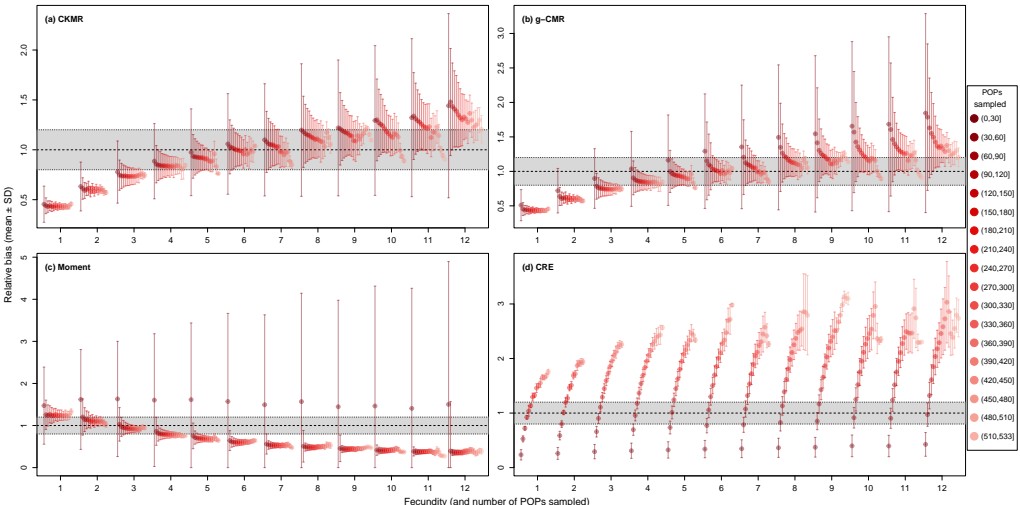

**Figure 5** **Relative bias ($\hat{N}/N$), mean ± SD) as a function of the fecundity (ranging from 1 to 12), and the number of parent–offspring pairs (POP) sampled for the (A) CKMR, (B) g-CMR, (C) Moment and (D) CRE methods.** CKMR and CRE methods estimate adult population size while g-CMR and Moment methods estimate the whole population size and the breeding female population size, respectively. The long-dashed horizontal line represents the optimal value for an unbiased estimator, estimators above this line are overestimating the true population size while estimators below are underestimating it. Points in the grey area represent estimations within 20% of the true population size. Results for fecundity > 12 are not shown to increase visibility (original figure can be found in Fig. S3).

and the biased estimates might mask actual population changes, making them unreliable indicators of population dynamics.

Fecundity appeared to be the most important factor for the CKMR, g-CMR and Moment methods' accuracy, with optimal fecundity values ranging from 4 to 6 offspring/female for the CKMR and g-CMR methods, and from 2 to 4 for the Moment method. This explains why, contrary to our hypothesis, CKMR and g-CMR wild boar estimations were more accurate than red deer estimations. The Moment method is thus more adapted to species with a low fecundity like cervids with litter size usually ranging from one to three offspring (*Jones et al., 2009*), while the CKMR and g-CMR methods are better suited for species such as the wild boar, or meso- to large carnivores such as foxes or wolves with larger litter size (*Jones et al., 2009*).

The CRE method was relatively insensitive to fecundity values and appeared to be the most precise method with a constant low CV (*i.e.,* <8%). However, given that its accuracy is highly dependent upon the sampling intensity, we recommend that it should be used only in areas where the population is small enough to be estimated roughly in advance, so that researchers can ensure that sampling achieved adequate coverage of the population (at least ~50%).

The good performance of the methods for long-lived species with high fecundity values is surprising, as is the insensitivity to the variation in fecundity. Indeed, one assumption of the basic versions of the four POPs-based methods applied here is that each individual in the population has the same probability of being captured, but this assumption is expected
to be violated to some degree for long-lived species. In polygynous mammals for example, competition between males for access to receptive females is assumed to lead to a variance in male mating success (*Clutton-Brock, 1989*), when older and larger males can monopolize groups of females, increasing their reproductive success (*e.g.*, red deer, *Carranza, Alvarez & Redondo, 1990*; *Clutton-Brock, Guinness & Albon, 1982*; wild boar, *Delgado et al., 2008*). This might increase the probability of sampling their offspring and bias the probability of recapture in their favor compared to young males. We tried to reproduce this type of bias by simulating populations with high variation in fecundity, but our results showed that all the methods seem to be robust to such bias.

Additional bias might come from the sampling strategy, as we simulated a random sampling of each population, which is likely not the case for hunted populations. Although introduced bias is routinely corrected for in fisheries systems, similar bias is often ignored in terrestrial systems (*Martınez et al., 2005*). Several factors might lead to biased hunting bag data toward a particular group of individuals such as hunting regulations (*e.g.*, the requirement to harvest animals according to certain quotas regarding sex or age), hunter preferences (*e.g.*, trophy hunting which targets animals with exceptional phenotypic traits, *Palazy et al., 2012*), hunting methods (*e.g.*, stalking *vs.* drives), or animal behaviour (*e.g.*, social *vs.* solitary species) (for a review see *Mysterud, 2011*). As a consequence, age composition and sex ratio of the bag does not always reflect the population composition (*Bunnefeld et al., 2009*). *Müller, Mercker & Brün (2020)* showed that g-CMR tends to strongly underestimate population size when parent–offspring pairs are harvested together by hunters. *Conn et al. (2020)* showed that the CKMR method is relatively robust to low dispersal level (*i.e.,* when related individuals tend to live spatially close to each other) and spatially biased sampling when simulating long-lived mammal populations with low rate of harvesting. In contrast, *Davies, Bravington & Thomson (2017)* showed that CKMR population estimations for Atlantic bluefin tuna (*Thunnus thynnus*) can be strongly biased when the spatial structure of the harvested population was not explicitly accounted for.

The POPs-based methods sensitivity to the different biases induced by hunting must thus be investigated more thoroughly, for example through spatially-explicit simulations, and if needed, these bias should be explicitly modelled by developing spatially structured estimators (*Conn et al., 2020*). We also believe that life-history data should be collected routinely during hunts, in addition to tissue samples, to allow adjusting the probabilities of capturing close kin based on sex, age or size, using a pseudo-likelihood approach for example as it is possible with the CKMR approach (*e.g.*, *Ruzzante et al., 2019*, Supplemental Information S3). Sex can be determined genetically, but it is more complex for age estimation. Recent methods using DNA methylation ($DNA_m$) have been proposed to estimate individuals' age for certain species (*De Paoli-Iseppi et al., 2017*), and have recently reached a sufficient level of accuracy to be used with POPs-based methods (*e.g.*, *Lemaître et al., 2022*; *Mayne et al., 2021*). They could be an interesting alternative for game species, but counting teeth cementum annuli currently remains the most used method (*e.g.*, *Pérez-Barbería et al., 2014*). Hunters are already involved in game species monitoring in several countries, for example, in some sites, moose Norwegian hunters are required to submit jawbones, ovaries and body weight information of the animals harvested (*Cretois*
*et al., 2020*). Such practices could be generalized as they do not involve much more field investment and could help improving population size estimation.

For this article, we assumed perfect pedigree reconstruction, but imprecision and errors in genetic-based relatedness estimates might decrease the accuracy of census size estimations. The ideal genetic markers for parent–offspring identification are high resolution genetic (microsatellites) or genomic (SNPs) to provide reliable relatedness estimates beyond first-order relationships to facilitate pedigree reconstruction (*Creel & Rosenblatt, 2013*). Several reviews discuss the range of genetic markers suitable for pedigree reconstruction, and appropriate methods of analysis (*Jones et al., 2010*; *Pemberton, 2008*; *Wang, 2019*; *Wang & Santure, 2009*). For example, *Riester, Stadler & Klemm (2009)* showed that the reconstruction of large and deep pedigrees can be accurate with only 10–15 polymorphic microsatellite loci, and *Ekblom et al. (2021)* showed that ~100 SNPs outperformed 19 microsatellites for the pedigree reconstruction of a wolverine (*Gulo gulo*) population. Recent rapid advancements in genetic and genomic approaches are making large panels of microsatellites or SNPs quickly obtainable for many species (*Meek & Larson, 2019*) and for a moderate price (*Waples & Feutry, 2022*). However, further studies should also include the effect of pedigree reconstruction uncertainties on the precision and accuracy of population size estimation, particularly applications of the CKMR using more distant relationships (*e.g.*, half-sibling pairs, *Hillary et al., 2018*).

## CONCLUSION

Recent studies have shown that current methods used to estimate population abundance of game species need to be improved to be more useful for management purposes (*Guerrasio et al., 2022*), particularly as hunted ungulates have reached unprecedented high densities in most of Europe (*Apollonio, Andersen & Putman, 2010*) leading to increasing human-wildlife conflicts (*e.g.*, *Massei, Roy & Bunting, 2011*). However, most current methods require months or years of intensive sampling, and applying them more rigorously would require a greater effort in terms of personnel involved. This is likely to be challenging to achieve, as the number of hunters and trappers, who often volunteer samples for population size estimation, is decreasing in some parts of Europe (*Massei et al., 2015*; *Riley et al., 2003*). In this context, POPs-based methods using hunting bag data represent a promising avenue as they do not require additional field investment and could be used as a core component in monitoring programs augmented with other methods. However, it is also important to understand their limitations, and our results show that—at least in their naïve form—they are only applicable under certain conditions. The minimum sampling intensity is a factor particularly important and unlikely to be met for some species without using multiple DNA sources. To ensure that the estimated population trends reflect true population dynamics, biases should be carefully investigated whenever possible. This could be achieved through more complex simulations mimicking potential biases created by hunting regulations or hunter preferences. Overall, while additional work is needed to fully understand the usefulness of POPs-based methods for estimating population sizes in terrestrial game species, our simulation results are promising, and suggest that the potential of these methods for applied wildlife management is high.

## ACKNOWLEDGEMENTS

We are very grateful to Johannes Signer for his insights on the simulations design, and to Scott Appleby for his comments.

### Funding

The authors received no funding for this work.

### Competing Interests

The authors declare there are no competing interests.

### Author Contributions

- Jeremy Larroque conceived and designed the experiments, performed the experiments, analyzed the data, prepared figures and/or tables, authored or reviewed drafts of the article, and approved the final draft.
- Niko Balkenhol conceived and designed the experiments, authored or reviewed drafts of the article, and approved the final draft.

### Data Availability

The R scripts to simulate population pedigrees and to estimate census population size are available in the Supplemental Files.

### Supplemental Information

Supplemental information for this article can be found online at http://dx.doi.org/10.7717/peerj.15151#supplemental-information.

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
