# Peer review of "A simulation-based evaluation of methods for estimating census population size of terrestrial game species from genetically-identified parent-offspring pairs"

_PeerJ, doi:10.7717/peerj.15151_

## Round 0.1 · original submission · Major Revisions

The reviewer's comments are appropriate and all should be considered when revising the manuscript. Particular issues include:

- One reviewer questions whether the work serves as a general comparison of CKMR with the other 3 close-kin approaches, given the applied CKMR methodology.

- A number of assumptions are made that require clarification and discussion, so that the simulations can be assessed in the appropriate comparative context.

- There are a number of grammatical issues and some instances of non-scientific language. Please revise for clarity and redundancy.

·

Basic reporting

All ok

Experimental design

I think the research questions could be better defined, see additional comments

Validity of the findings

All ok

Additional comments

The study “A simulation-based evaluation of methods for estimating census population size of terrestrial game species from genetically-identified close-kin” uses custom script to generate small populations with traceable parent-offspring pairs in order to assess 4 different methods to estimate population size.

I believe there is value in this study, but I think in its current state there is a risk to mislead readers as to what these methods may be able to achieve for their particular species of interest. I am familiar with CKMR but not the other 3 methods and my comments therefore mostly relates to the CKMR method and its implementation here. The authors reduced CKMR to its very simplistic “cartoon” version which doesn’t take into account the age of the individuals involved in POPs (or any other life history parameters). Each POP provides information about the size of the adult population when the offspring was born and the adult population size needs to be integrated over time. In that context, overlapping generations are not an issue.
To me, these simulations answer the following question: can CKMR still be useful even if used in its very simplistic version, without knowing anything about the life history of a species or the age of the individuals sampled. I am not sure whether similar shortcuts were taken with the other methods explored here, but I don’t think this study qualifies as a general comparison of (at least) CKMR with the other 3 close-kin approaches. I do think there is value in the study done here, but I think it’s currently misleading the readers as to what that comparison really is. The title and the abstract illustrate this fairly clearly:
- in the title, “of terrestrial game species” really should be replaced by “in the absence of any information about life history and age” (I am not suggesting this as a title, just trying to make a point)
- the abstract reads like the main issue with current methods is overlapping generations, when I believe the main issue is the potential lack of information about life history/age for game species

This is really is an evalution of the kin methods under worst-case scenarios (again, at least for CKMR), which I think needs to be highlighted through the manuscript (title, abstract, introduction with defined objectives and in the discussion). In order to do that appropriately it would be good to detail the assumptions and/or changes made in the simulations for this study compared to “normal use” scenarios for each of the 4 methods evaluated here, maybe as a table. Currently there are all dumped as “assume non-overlapping generations” and some words in paragraph L106-115 for all 4 methods without any distinction: e.g. is age important for CKMR, CRE, Moment estimator and g-CMR? Can they correct for deviations from underlying assumptions?
Once it is clear through the manuscript that the evaluation is really for worst-case scenario and doesn’t inform about the true potential of these methods, I think it would represent a interesting contribution to the field. In essence this work highlights the robustness of (some) of these methods to violation of various assumptions and their potential as a monitoring tool, possibly to follow the population trend rather than its exact size. It should also probably be noted somewhere that whilst the goal is to estimate census size, what is in fact estimated leans towards Nb, the effective population breeding size.

Additional comments

L3: I think the title should replace “close-kin” with “parent-offspring” since no other kin types are used to improve the estimate of census population size.
L71: CKMR can incorporate all sort of life history information, not just fecundity into the pseudo-likelihood framework
L79: there are more CKMR studies, see Bravington et al. 2019 (https://www.nespmarine.edu.au/document/close-kin-mark-recapture-population-size-estimate-glyphis-garricki-northern-territory), Bradford et al. 2018 (https://www.nespmarine.edu.au/document/close-kin-mark-recapture-estimate-population-size-and-trend-east-coast-grey-nurse-shark), Thomson et al. 2020 (https://www.frdc.com.au/sites/default/files/products/2014-024-DLD.pdf),
Delaval et al. 2022 (https://onlinelibrary.wiley.com/doi/full/10.1111/eva.13474)

L96-105: this is a non-issue for the CKMR pseudo-likelihood framework that integrates population size over time. As discussed in my general comments, the main issue is more about 1) the lack of age information to define when the offspring was born and integrate population size over time; 2) the lack of information about life history which is partly covered in the next paragraph
L106-115: I understand smaller sample sizes are generally available for terrestrial game species, but if a kin-based study was to be carried out, sample sizes would have to be relatively large (as demonstrated by this study) and some life history information could be collected then. Also, probably worth mentioning in the discussion, sex or age information could potentially be obtained via a genetic approach
L126-128 and L275-277: for CKMR at least, variable fecundity is not an issue if random for each new breeding event, it only becomes an issue if some individuals consistently have higher/lower reproductive outputs than others (see Waples et Feutry 2022). It is not clear to me from this paragraph or the methods section how the variability was implemented, was it consistently lower/higher over the lifetime of an individual or was it redefined for each breeding event?

Reviewer 2 ·

Basic reporting

Thank you for the opportunity to review your manuscript “A simulation-based evaluation of methods for estimating census population size of terrestrial game species from genetically-identified close-kin (#78485)” in PeerJ. In this manuscript, the authors explore four estimators of population size by reconstructing a perfect pedigree of red deer and wild boar. The authors found that estimating population size using all models is possible in both species, but recommend caution given sampling density, species-specific fecundity, and other biases from how individuals are sampled.

I think this is an important concept that is worth consideration wildlife management of harvested species, and beyond. I believe the authors did an excellent job reviewing methods that are available for estimating population size from subsampled pedigree data. I especially enjoyed Box 1, and its ability to effectively communicate the four methods for estimating population size from kinship.

There are some areas where the manuscript would benefit from more clarity. For example, in the abstract, the authors may consider including information on the systems being studied (e.g., wild boar and red deer), to provide important context to the readers. Further, the Introduction can benefit from a broader information on the need for census population sizes in ecology and conservation, to extend the applicability of this study.

One major component missing from this manuscript is discussion regarding how genetic tools are incorporated into estimating relatedness and generating pedigrees for these analyses. At present, the Authors generate complete pedigrees that can show kinship between individuals. However, generating pedigrees of wild individuals from genetic data can be difficult (Pemberton 2008), and depending on the markers used (e.g., microsatellites, SNPs; Galla et al. 2020) and the sampling density (Wang 2016), there can be ambiguity or imprecision in relatedness estimates and pedigrees generated from genetic data. A rigorous discussion of the types of markers used, the caveats, and costs, should be incorporated into the Introduction and Discussion.

Finally, I've provided a few specific comments that may help strengthen clarity throughout the manuscript:

Lines 36-40 (and throughout the manuscript): For clarity of reading, put the i.e., and e.g., comments in parentheses.

Line 53: I would consider removing the word ‘basically’ here, to make the sentence cleaner.

Lines 52-60: I enjoy the descriptions of each method, provided by the authors. I would consider referencing Box 1A and Box 1B when referring to the mark-recapture and kinship methods.

Line 83: You can remove “i.e.,” from this sentence, and it will read much better. I encourage the authors to check their use of i.e., and e.g., throughout this manuscript to ensure they are necessary.

Lines 92-95: There is an important step missing from the Introduction, and that is how relatedness is estimated or pedigrees are reconstructed from genetic data. This will be helpful for the readers to understand how to get from genetics to kinship.

Line 205: Typo with ‘bellow’.

Lines 242-247: The authors could include other ways that people could acquire tissue samples outside of hunting. For example, collecting DNA from fecal samples (e.g., Pigroni et al. 2006) or snares (e.g., Gardner et al. 2010) might be a good method for acquiring a higher sample size.

References Cited:

Galla, S. J., Moraga, R., Brown, L., Cleland, S., Hoeppner, M. P., Maloney, R. F., ... & Steeves, T. E. (2020). A comparison of pedigree, genetic and genomic estimates of relatedness for informing pairing decisions in two critically endangered birds: Implications for conservation breeding programmes worldwide. Evolutionary Applications, 13(5), 991-1008.

Gardner, B., Royle, J. A., Wegan, M. T., Rainbolt, R. E., & Curtis, P. D. (2010). Estimating black bear density using DNA data from hair snares. The Journal of Wildlife Management, 74(2), 318-325.

Pemberton, J. M. (2008). Wild pedigrees: the way forward. Proceedings of the Royal Society B: Biological Sciences, 275(1635), 613-621.

Prigioni, C., Remonti, L., Balestrieri, A., Sgrosso, S., Priore, G., Mucci, N., & Randi, E. (2006). Estimation of European otter (Lutra lutra) population size by fecal DNA typing in southern Italy. Journal of Mammalogy, 87(5), 855-858.

Wang, J. (2017). Estimating pairwise relatedness in a small sample of individuals. Heredity, 119(5), 302-313.

Experimental design

As mentioned previously, incorporating pedigree uncertainty into the simulation would provide an important consideration for people estimating population size using different marker types. I appreciate this may be outside of the scope of this study, but should be described in the Introduction and Discussion as an important consideration nonetheless.

Validity of the findings

I appreciate the authors created a robust simulation with many replications that is reproducible given the R scripts provided in the supplemental. I have no further notes regarding the validity of the findings.

As mentioned previously, a robust discussion of how pedigrees and relatedness are estimated is needed in the Discussion. While the authors find that 40% of a population needs to be sampled to provide robust estimates or population size, this may not be achievable given most conservation budgets (given the cost of genetic/genomic sequencing). This is worth discussion in the Conclusion, as well.

Additional comments

None.

Reviewer 3 ·

Basic reporting

no comment

Experimental design

Line 98-100: Here I cannot agree unreservedly. If, for example, an adult (i.e. 2 years or older) female or male wild boar is shot and sampled in 2020, the relationships to a subadult wild boar shot in 2021 can certainly be reconstructed retroactively. It should therefore be possible to adjust the input data for the different estimators accordingly. Whether and to what extent this leads to a bias (lines 104-105) would have to be tested separately.

Line 114-115: Are these data really so rare? I would have assumed that for both single and drive hunts, at least sex, age and dressed weight are determined. And the sex of a tissue sample can still be determined genetically afterwards. Especially for Germany, such statistics should be available in the context of the hunting bags (e.g. see Keuling et al 2010 and 2013).
At the same time, I wonder if the smaller sample sizes for terrestrial game species are really a problem? After all, the populations are smaller than those of farmed fish. Especially for red deer and wild boar, the hunting numbers of the Central European countries France, Germany, Poland and Italy are large

Line 145: Why 3 years? It is well known that weight, not age, is the determining factor for reaching sexual maturity (Frauendorf et al 2016, Drimaj et al 2019, Gayet et al 2021). We increasingly observe pregnant piglets (Gethöffer et al 2007) and males with mature sperm from a body weight of 30 kg. Accordingly, a large proportion of males in a population must at least be considered as putative fathers. I wonder therefore if 3 years is not too high an age?

Line 146-147: Here I would like to understand why you chose the 44%. I am aware that Gayet et al 2016 even speaks of up to 60%. Other studies (Costa et al., 2012; Say et al., 2012; Müller et al 2018) have values between 20% and 50%. In principle, I would assume that the size of this value, e.g. 30% or 40%, has only a small influence on the simulation results; to me, the 44% used simply seems a little too high for an average European wild boar population.

Validity of the findings

Line 272-277: More of a comment here than a suggestion for improvement. Keuling et al., for example, were able to show that for more than a decade, piglets in particular, but also subadults, have been the motor of population growth in wild boar. For Central Europe in particular, it can no longer be assumed that adult males still play a determining role in the mating systems of the populations. On the one hand, this is due to the fact that old boars are becoming increasingly rare (Gayet et al 2021 show for e.g. northern France that there are no longer populations in which male wild boar are moderately hunted or even unhunted.) and, on the other hand, the competitive pressure with subadult boars has become far too great to monopolise entire groups of females. The probability of successful subadult sneakers can be considered quite high. On the other hand, this could also explain the sometimes quite high number of multiple paternities. A bias, against which the models seem to be robust in any case, is therefore probably not present. Accordingly, you might consider rephrasing the text.

Additional comments

no comment

Annotated reviews are not available for download in order to protect the identity of reviewers who chose to remain anonymous.

---

## Round 0.2 · Minor Revisions

Dear author,

Thank you for submitting your revision. You have nicely addressed and captured the comments and concerns of the reviewers and I and there are few remaining edits required (and those are extremely minor).

I invite you to consider and address the small number of comments from the reviewers. For myself, I would echo one reviewer who called for greater consideration of recent developments and publications in population genetic studies.

Best wishes,
Anthony

·

Basic reporting

All good

Experimental design

All good

Validity of the findings

All good

Additional comments

I think in general the authors did a good job at addressing the comments raised by both reviewers and I think the paper is now up to publication standards of PeerJ. Following this revision I do have a few additional comments that may help improve the the manuscript a bit further:
- given parent-offspring pairs (POP) are the only relationship used in this study, I don't think the genetics to find them are a big deal. Things have improved a LOT since 2008 and POPs can be found with extremely high accuracy at a moderate cost (less than $10 a sample, see Waples & Feutry 2022 for some info on pricing for CKMR). Also microsatellite vs SNP is no longer a debate. I would therefore probably reduce that section and stick to references from the last 5 years.
- worth having a sentence stating that more distant relationships can and have been used (see Hillary et al. 2018 already cited and Patterson et al. 2022) in which case more care is required to find the related pairs and dealing with uncertainty becomes a real thing
- DNA ageing is actually working really well, much progress has been made since that study from De Paoli, as an example we are currently replacing all our otolith aging in fish with epigenetic aging, see Mayne et al. 2021 for a recent reference
- worth adding Patterson et al 2022 to the list of recent CKMR developments line 209

References
Patterson, T. A., Hillary, R. M., Kyne, P. M., Pillans, R. D., Gunasekera, R. M., Marthick, J. R., ... & Feutry, P. (2022). Rapid assessment of adult abundance and demographic connectivity from juvenile kin pairs in a critically endangered species. Science Advances, 8(51), eadd1679.

Mayne, B., Espinoza, T., Roberts, D., Butler, G. L., Brooks, S., Korbie, D., & Jarman, S. (2021). Nonlethal age estimation of three threatened fish species using DNA methylation: Australian lungfish, Murray cod and Mary River cod. Molecular Ecology Resources, 21(7), 2324-2332.

Reviewer 2 ·

Basic reporting

Thank you for the opportunity to re-review your manuscript “A simulation-based evaluation of methods for estimating census population size of terrestrial game species from genetically-identified close-kin (#78485)” in PeerJ. I believe this version is a great improvement over the original. I found it easier to read, with more context that is important for the readers. I only have one comment to strengthen the Introduction:

Lines 56-67: As mentioned in the first review, having more explicit information here about the genetic tools used generate an understanding of ‘kin’ are important to highlight early. I recommend putting 1-2 sentences in this paragraph on what the genetic data look like (i.e., using microsatellite markers or SNPs depending on the study system at hand) and what estimators are used to generate kinship (e.g., identity-by-state). This is a simple fix, but provides important information early for practitioners who are reading this information.

Experimental design

No comment.

Validity of the findings

No comment.

Additional comments

I have a few minor suggestions to improve clarity of the manuscript:

Line 91: Replace “seems to be” with “is currently the most utilized”

Line 138: Change “pedigrees” to “pedigree”.

Line 156: “15 years” instead of “15 year”

Line 334: Change to “but imprecision and errors in genetic-based relatedness estimates…”

Line 335: Change to “The ideal genetic markers for parent-offspring identification are high resolution genetic (microsatellite) or genomic (single-nucleotide polymorphisms) to provide reliable relatedness….”

Reviewer 3 ·

Basic reporting

No further or new comments.

Experimental design

No further or new comments.

Validity of the findings

No further or new comments.

Additional comments

In my opinion, the comments of the different reviewers have been satisfactorily implemented by the authors Larroque and Balkenhol. Not all comments have been included in the revised manuscript, but they have been sufficiently explained in the response section. This at least concerns my part!
The revised MS reads smoothly and stringently. From my point of view, there are no further questions or comments on the MS in its present form.

---

## Round 0.3 · accepted · Accept

Dear author,

Thank you for your resubmission and your efforts in addressing the reviewers' comments throughout this review process. I am happy to recommend publication of your manuscript.

Best,
Anthony